# Optical information transmission through complex scattering media with optical-channel-based intensity streaming

Haowen Ruan [1,2✉], Jian Xu [1,2✉] & Changhuei Yang [1✉]

For the past decade, optical wavefront shaping has been the standard technique to control light through scattering media. Implicit in this dominance is the assumption that manipulating optical interference is a necessity for optical control through scattering media. In this paper, we challenge this assumption by reporting on an alternate approach for light control through a disordered scattering medium – optical-channel-based intensity streaming (OCIS). Instead of actively tuning the interference between the optical paths via wavefront shaping, OCIS controls light and transmits information through scattering media through linear intensity operations. We demonstrate a set of OCIS experiments that connect to some wavefront shaping implementations, i.e. iterative wavefront optimization, digital optical phase conjugation, image transmission through transmission matrix, and direct imaging through scattering media. We experimentally created focus patterns through scattering media on a sub-millisecond time-scale. We also demonstrate that OCIS enables a scattering medium mediated secure optical communication application.

[1] Department of Electrical Engineering, California Institute of Technology, Pasadena, CA, USA. [2] These authors contributed equally: Haowen Ruan, Jian Xu.
✉email: hruan@caltech.edu; jxxu@caltech.edu; chyang@caltech.edu

Seeing through the fog, looking around corners, and peering deep into biological tissue have traditionally been considered to be impossible tasks in optics. The main challenge is attributable to disordered optical scattering that scrambles the optical field of light from different optical paths. In the last decade, optical wavefront shaping has made great progress in controlling light transmission through complex disordered scattering media for imaging and focusing[1–6]. This class of techniques first characterizes the optical phase or complex field of light from different scattering paths and then actively manipulates the output field by shaping an input wavefront. This active control of optical wavefronts has become a powerful and standard technique to manipulate light through disordered scattering media.

While different versions of optical wavefront shaping techniques have been developed, they almost always require active interference control and spatial light modulation[1–6]. During the measurement process, interferometry of various forms is used to measure the phase relationship between scattering paths. Then, knowledge of the phase information is used by a wavefront-shaping device to spatially modulate the phase or amplitude of a light field on the input side so that a desired optical pattern is obtained through the scattering medium by interference between all of the modulated optical paths. It is important to note that regardless of whether amplitude or phase modulation is used, traditional wavefront-shaping techniques always exploit the phase relationship between different optical channels to form the desired output pattern since the output pattern is formed by interference.

This principle means that optical wavefront shaping requires knowledge of the phase relationship between the input and output planes of the scattering medium, whether implicitly or explicitly. Mathematically, the optical fields on the input and the output plane are related by a complex matrix called the transmission matrix[5,7]. Since a typical transmission matrix can contain millions of entries or more, directly measuring both the real and imaginary parts of all the entries is a challenging task. Because of this challenge, significant research efforts have been devoted to improving the speed and stability of complex transmission matrix characterization[8,9].

Given the complexities associated with measuring the phase relationship between different channels for wavefront shaping, it is worth asking whether active interference control is a prerequisite for manipulating light through scattering media. Here, we report an optical approach that allows us to manipulate light through complex media without knowledge of the complex transmission matrix or the use of wavefront shaping. This method characterizes the optical intensity channels of the scattering medium by simply measuring the intensity of the optical speckle pattern transmitted through the scattering medium. Once we obtain the map of the optical intensity channels, we can stream photons through these optical intensity channels. Since the phase relationship between these channels is not measured, a spatially incoherent light source or temporal separation of the coherent light transmissions is used to achieve linear superposition of the photons from different channels. We call this method optical-channel-based intensity streaming (OCIS). This concept expands our understanding of light control through scattering media and introduces alternative strategies to overcome and use optical scattering.

In this paper, we first explain the concept of optical intensity channels and the approach for performing incoherent and linear operations with them. This concept is further generalized with an intensity transmission matrix framework based on the transmission matrix theory from wavefront shaping. We then experimentally demonstrate the ability of this method to form a focus pattern with feedback-based OCIS. We derive the

relationship between the number of controllable modes and the contrast-to-noise ratio (CNR) of the focus pattern. Interestingly, OCIS is also able to form an energy-null spot, a function that is difficult to achieve with wavefront-shaping approaches due to the presence of naturally occurring dark speckles[10,11]. We then move on to report on the optical intensity transpose, an OCIS-derived technique that uses optical speckle intensity information transmitted from a point source through a scattering medium to identify the optical channels of the scattering medium and to send light back to the location of the point source through these optical channels. Finally, using the linear operations provided by the intensity channels, we demonstrate a practical application of OCIS—scattering medium-mediated secure optical information transmission. OCIS can provide a means to transmit information in a secure way without requiring a prior secure channel.

## Results

**Principles**. To help understand the principles of OCIS for optical control through scattering media, let us imagine a scenario where a coherent light beam from a point source at position P1 is incident on and transmits light through a scattering medium (Fig. 1a). A laser speckle pattern will develop behind the scattering medium as a result of the mutual interference of multiple scattering paths[12]. The speckle intensity is randomly distributed, and let us assume that a bright speckle is developed at position P2 and a dark speckle is developed at position P3 (Fig. 1a). Whether a speckle is bright or dark depends on the degree of alignment (i.e., the degree of constructive or destructive interference) between the phasors representing the field contribution from different optical paths through the medium. The bright speckles result from situations where the phasors are more strongly aligned than on average (i.e., relatively constructive interference), forming a longer-than-average resultant phasor, and the dark speckles from the situations where the phasors are more weakly

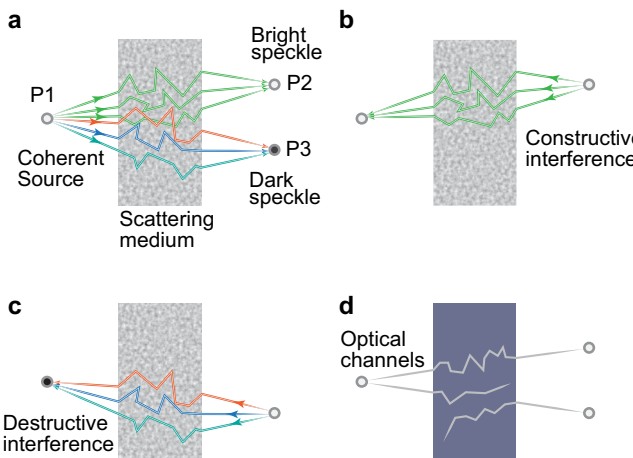

**Fig. 1 Optical intensity channels. a** A coherent source or Guidestar at position P1 on the input plane leads to a bright speckle at position P2 and a dark speckle at position P3 on the target plane. In the case of a bright speckle, the majority of the optical paths share a similar phase (denoted by the same color) and thus relatively constructively interfere. In contrast, the optical paths leading to the dark speckle are out of phase (denoted by different colors). **b, c** Based on the time-reversal symmetry of optical propagation, a bright speckle at position P2 will also lead to a bright speckle at position P1 (**b**). However, a bright speckle at position P3 will not result in a bright speckle at position P1 due to destructive interference (**c**). **d** This phenomenon is interpreted in a picture of optical channels. A 'bright' optical channel is established between position P1 and P2 for optical energy transmission while the optical channel between position P1 and P3 is 'dark'.

aligned than on average (i.e., relatively destructive interference), forming a shorter-than-average resultant phasor.

In the example presented in Fig. 1a, the relatively constructive interference occurring between light from the majority of optical paths that connect positions P1 and P2 forms a bright speckle at P2, whereas the relatively destructive interference of light occurring between the different paths that connect P1 and P3 forms a dark speckle. The time-reversal symmetry of optical propagation also means that if we place the light source at position P2, the light will follow the same trajectories to reach position P1 (Fig. 1b). Since the phase relation between these optical paths is maintained, relatively constructive interference occurs at position P1. Similarly, if we move the light source to position P3, destructive interference occurs at position P1 because the phase relationship between the optical paths remains the same, regardless of the propagation direction of the light (Fig. 1c). We can abstract the above analysis to a picture of optical channels (Fig. 1d). A 'bright' optical channel is established between the positions P1 and P2, while the optical channel connects position P1 and P3 is 'dark'. We can then generalize this relationship to the entire target plane where the intensity of a speckle maps to the throughput of the associated channels. The concept of optical intensity channels forms the foundation of OCIS, which allows us to manipulate light through scattering media by learning and modulating the intensity of light through the optical channels.

It should be noted that the concept of optical intensity channels here differs from the channels described in optical wavefront-shaping theory[13] since here knowledge of optical intensity alone is required, instead of information about the complex field. Mathematically, we can describe OCIS with an intensity transmission matrix, which was also used by Boniface et al.[14] recently. This mathematical theory mirrors the complex optical field transmission matrix theory that is extensively used in wavefront-shaping methods[5,7].

To understand OCIS in the framework of an intensity transmission matrix, we start by examining the complex optical field transmission matrix mathematical framework[5,7]. In this case, the optical fields on the input and the target plane can be discretized into complex row vectors $\mathbf{u}$ and $\mathbf{v}$, respectively, and connected by a transmission matrix $\mathbf{T}$ through the equation $\mathbf{v} = \mathbf{uT}$. The optical field on the target plane is a linear transform of the field on the input plane. However, the intensity on the target plane, which is of interest in most applications, is not linear with respect to the intensity of the shaped optical field.

OCIS aims to simplify this nonlinear relationship by directly connecting the input intensity to the output intensity in a linear form through an intensity transmission matrix $\mathbf{S}$ of the form

$$\mathbf{b} = \mathbf{aS} \qquad (1)$$

where $\mathbf{a}$ and $\mathbf{b}$ are row vectors denoting the intensity of the optical patterns on the input and output plane, respectively. Each element of $\mathbf{a}$ and $\mathbf{b}$ represents the intensity value of an optical mode. This equation is valid if the input modes are spatially incoherent with each other so that the intensity of each output mode is a linear combination of the intensity values of these input modes. In this case, The $(i,j)$th element of $\mathbf{S}$ is equal to the magnitude square of the $(i,j)$th element of $\mathbf{T}$, and thus all elements of $\mathbf{S}$ are real and non-negative. While complex transmission matrix theory interprets light propagation through scattering media on a fundamental level, the intensity transmission here serves as an intuitive and efficient tool to analyze linear and incoherent operations.

There are two primary ways by which we can satisfy the condition that the spatial modes on the input plane are spatially incoherent and do not mutually interfere with each other. The most direct way is to use a spatially incoherent light source on the input side. Alternatively, we can sequentially illuminate each input spatial mode. In this case, time separation can also guarantee that cross-modal interference does not occur. In our experiment, we demonstrated the use of these two approaches to linearly and incoherently operate on the intensity channels. In the following sections, we will demonstrate a series of experimental implementations of OCIS to overcome optical scattering and transmit information through disordered scattering media.

**Feedback-based OCIS.** Here we apply the principle of OCIS with a feedback mechanism to form a focus pattern through a scattering medium, an important evaluation of the ability of this technique to overcome optical scattering. The requirement for a feedback mechanism here shares similarity with feedback-based wavefront shaping[15]. The implementation of feedback-based OCIS can be divided into two steps, measurement and display. During the measurement process, OCIS aims to find the bright and dark optical channels between the input plane and the target spot. In this case, one can use a CW laser source to illuminate the scattering medium with different optical modes, e.g., scanning spatially over time as shown in Fig. 2a. By simply measuring the optical intensity feedback as the laser beam scans, one can learn the optical channel mapping between the scanning position on the input plane and the target spot. During display, we simply inject light only into the optical channels that connect the input plane and the target spot as shown in Fig. 2b. Although part of the light also couples to other channels that are connected to other positions on the target plane, the total light intensity on these positions is on average lower than that on the target spot. It should be noted that in addition to position scanning, other approaches such as angular scanning and wavelength sweep are also able to excite different channels of a scattering medium.

We now use the intensity transmission framework to analyze feedback-based OCIS. During the measurement step, we sequentially send in the basis input vectors, which form a matrix $\mathbf{A}$ (Fig. 2c, left matrix). In our case, we input single modes (i.e., $\mathbf{A}_i = \boldsymbol{\delta}[i]$), where $\mathbf{A}_i$ denotes the $i$th row vector of $\mathbf{A}$, and $\boldsymbol{\delta}[i]$ is a delta row vector with a nonzero value at the $i$th element (e.g., $\mathbf{A}_1 = [1,0,0,\ldots]$). As a result, by collecting the transmitted intensity patterns over time, we obtain an output matrix $\mathbf{B}$ (Fig. 2c, right matrix) that maps to the intensity transmission matrix $\mathbf{S}$. The time-encoded intensity measured at the $j$th location on the target plane indicate the throughput of the optical channels that connect the corresponding input mode to this location.

To enhance the temporal average intensity at the $j$th location on the target plane, we select a subset of the row vectors of $\mathbf{A}$ where each row vector $\mathbf{A}_i$ connects to a high-throughput (bright) channel to the $j$th column of $\mathbf{B}$. We denote the set of row vector indices obtained using the feedback-based OCIS as $\mathbf{C}_{\mathrm{FB}}$. We then sequentially send light to these channels and integrate the output intensity patterns (Fig. 2d) to avoid mutual interference between different channels. Mathematically, the target pattern formed by feedback-based OCIS is given by

$$\mathbf{b}_{\mathrm{FB}} = \sum_{i \in \mathbf{C}_{\mathrm{FB}}} \mathbf{B}_i = \sum_{i \in \mathbf{C}_{\mathrm{FB}}} (\mathbf{A}_i \mathbf{S}) \qquad (2)$$

Eq. (2) can be further rewritten as

$$\mathbf{b}_{\mathrm{FB}} = \left( \sum_{i \in \mathbf{C}_{FB}} \mathbf{A}_i \right) \mathbf{S} = \mathbf{a}_{\mathrm{FB}} \mathbf{S} \qquad (3)$$

where $\mathbf{a}_{\mathrm{FB}}$ is the summation of the input mode intensities in time. This equation is in agreement with Eq. (1), which justifies the use of temporal separation approach to realize the requirement of spatial incoherence.

With this mathematical framework in place, we can quantitatively evaluate the performance of OCIS based on speckle

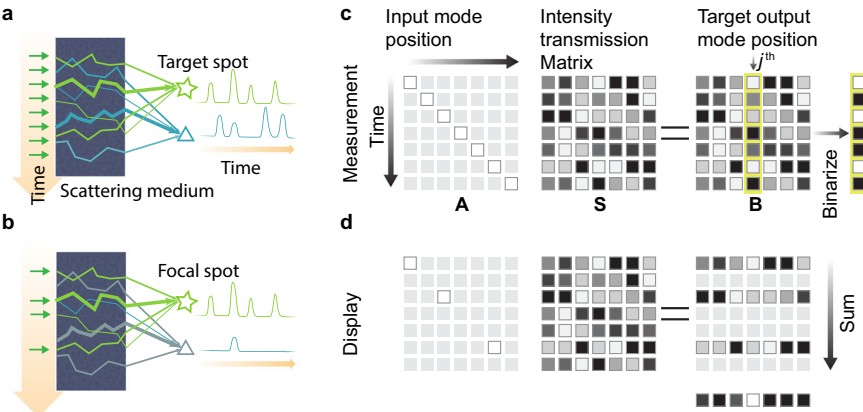

**Fig. 2 Principle of feedback-based OCIS. a** Coherent light source scans across the input plane of the scattering medium over time, resulting in time-varying intensity signals at the target spot on the target plane. At another spot (denoted by a triangle), the signals are uncorrelated with those at the target spot due to the random scattering of the light through the sample. These time-varying signals allow us to map the optical channels between the input and the target plane. **b** By injecting light into the channels that connect to the target spot, a focus pattern is formed at the target spot while other spots on the target plane receive less light on average. **c, d** Matrix representation of the feedback-based OCIS. **c** The incident optical mode sweeps through space over time, and can be represented by an identity matrix **A**. Its interaction with the scattering medium is represented by multiplying the intensity transmission matrix **S**, resulting in time-varying speckle patterns on the output. The measurement of the time-varying speckles at the target position is equivalent to taking one column of the intensity matrix **B**. **d** During display, the binarized output selects a number of rows of **S** as output. The integration of the selected output rows over time results in an optical focus pattern at the target position.

statistics. The contrast-to-noise ratio (CNR), which is defined as the ratio between the background-subtracted, time-averaged, spatial-peak intensity, and the standard deviation of the background, is a reasonable metric because it evaluates both the peak intensity of the temporal average pattern and the fluctuation of its background. Assuming the instantaneous speckle patterns, i.e., the rows of the intensity transmission matrix are fully developed[12], and the total number of uncorrelated speckle patterns that the OCIS system can measure and control is $N$ (i.e., the number of rows in **S**), the expected CNR of the optical spot pattern is given by

$$\mathrm{CNR} = \sqrt{N} \exp\left(-\frac{I_t}{2\mu}\right)\frac{I_t}{\mu} \qquad (4)$$

where $I_t$ is the intensity threshold and $\mu$ is the mean intensity of the speckle (i.e., the mean of the intensity transmission matrix **S**). A step-by-step derivation of Eq. (4) is provided in Supplementary Note 1. As the total number of measured frame $N$ increases, the background becomes more uniform, the CNR increases, and the resulting optical spot becomes more pronounced. Therefore, this metric indicates the ability of OCIS to overcome optical scattering and to recover optical information through scattering media.

Another metric that is widely used in optical wavefront shaping is the peak-to-background ratio (PBR) or intensity enhancement factor, which is defined as the ratio between the peak intensity and the mean of the background. The PBR of OCIS is given by

$$\mathrm{PBR} = 1 + \frac{I_t}{\mu} \qquad (5)$$

A detailed derivation of Eq. (5) is included in Supplementary Note 1. As shown in Eqs. (4) and (5), we can choose the intensity threshold $I_t$ to optimize either CNR or PBR (see Supplementary Note 1).

In wavefront shaping for optical focusing through scattering media, the PBR and CNR of the focus are equal, except for a constant offset of 1 (PBR = CNR + 1), for fully developed background speckle patterns. This fixed relationship stems from the fact that the background follows well-defined speckle statistics, where the mean and standard deviation of the background are the same value. In comparison, the PBR and

CNR are quite different quantities in OCIS because the background mean is decoupled from its variance. A more detailed discussion of CNR and PBR can be found in Supplementary Fig. 1. Different with wavefront shaping, both CNR and PBR are required here in OCIS to comprehensively characterize the quality of the focus pattern. CNR indicates the peak value and background variance, which determines the visibility of the focus pattern, while PBR indicates the energy enhancement on the targeted optical spot. For OCIS, CNR provides a better gauge of the signal-to-noise ratio than PBR in strong light scenarios (see Supplementary Note 1 for more details). Since most of our experiments were performed at high light-intensity levels, we chose to optimize CNR instead of PBR for optimal performance. We do note that in low-light scenarios, PBR becomes the more relevant gauge of the signal-to-noise ratio for OCIS.

We next report our experimental findings on the controlling capability and speed of feedback-based OCIS. A simplified signal diagram is shown in Fig. 3a, b, and the detailed experimental setup is described in the "Methods" section and shown in Supplementary Fig. 2a. During measurement, a CW mode laser source illuminates the scattering medium (a ground glass diffuser, see Methods). We use a single photodetector with an active area comparable to the size of a single-speckle grain to measure the temporal speckle intensity of the target point at the target plane during one galvo mirror sweep of time duration $t$ (Fig. 3a). We can then apply an intensity threshold to the time trace and identify a subset of optical channels that contribute bright speckles at the target location. During the display step (Fig. 3b), we use this information to selectively turn on the laser illumination only at time points when this subset of speckle patterns is reproduced during a repeated galvo mirror scan. Since all the selected speckle patterns show a brighter-than-threshold speckle at the target point, the temporal average optical intensity at the target point can then be expected to be higher on average than that of the background. Although the instantaneous intensity may fluctuate within the time period of $t$, i.e., the galvo mirror single-trip scan duration, the temporally averaged optical spot can, nevertheless, effectively fulfill the role of a wavefront-shaping-based focused spot in many applications such as imaging

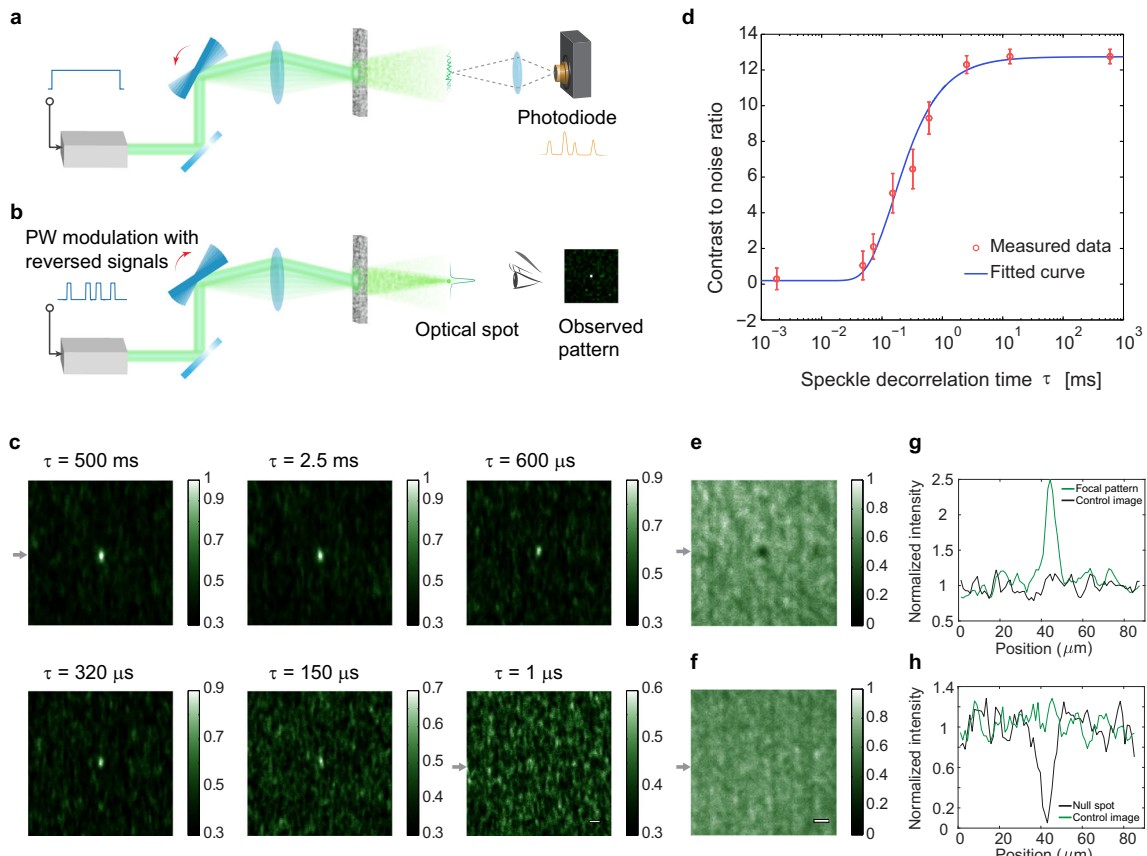

**Fig. 3 Results of the feedback-based OCIS. a** Simplified system setup for measurement. A galvo mirror was used to steer light into different channels of the scattering sample. As the galvo mirror scanned forward, the photodetector measured the temporal signal that fluctuated as light (continuous wave) coupled into different channels. **b** The measured signal was binarized and used to modulate the intensity of the laser in a time-reversed order as the galvo mirror scanned backward. In this case, we can measure the optical channels and inject light (PW, pulsed wave) into the bright channels during a galvo mirror round trip. **c** At different sample decorrelation times, optical spots were created in free space and captured by a camera with an exposure time of 125 μs. The CNRs for τ from 500 ms to 1 μs are 12.6, 12.3, 9.3, 6.5, 5.1, and 0.3. Scale bar: 20 μm. **d** CNR as a function of the sample decorrelation time. The error bar indicates the standard deviation of four points. **e, f** Feedback-based OCIS for null energy display. **e** By injecting light into the dark optical channels instead of the bright ones, we can obtain a null energy spot on the target plane. **f** Control image. By sending light into randomly selected channels, no null energy spot was observed. Scale bar: 20 μm. **g** Line plots of the arrow-indicated lines in (**c**, τ = 500 ms and 1 μs), normalized by the mean value of the background. **h** Line plots of the arrow-indicated lines in (**e, f**), normalized by the mean value of the background.

or target localization, as long as $t$ is shorter than (1) the decorrelation time of the scattering medium and (2) the application's signal integration time frame.

We used a comparator circuit to identify the high-intensity speckles measured by the photodiode and streamed the digital signal (Supplementary Figs. 3 and 4) to a field-programmable gate array (FPGA) that was synchronized with the galvo mirror. During the display process, a last-in-first-out module in the FPGA was used to time-reversed and output the signal as the galvo mirror swept back. In this case, the output signal from the FPGA modulated the laser. In our experiment, a galvo mirror of 4 kHz was used, meaning that an optical spot was created within 250 μs. This all-hardware-based OCIS system is able to measure and control $8 \times 10^3$ modes per millisecond (see Supplementary Note 2 for detailed analysis).

A camera with exposure time covering a galvo mirror one-way sweep (125 μs) was triggered to capture the patterns on the target plane. Figure 3c shows the patterns captured by the camera at various speckle decorrelation times. To demonstrate the performance of this technique in the presence of fast speckle decorrelation, we tuned the speckle decorrelation time by moving the scattering medium at controllable speeds. As shown in Fig. 3c, the visibility of the optical spot becomes lower as the speckle

decorrelation time decreases. To quantify the CNR as a function of speckle decorrelation time, we calculated the CNR of the patterns and plotted them over the decorrelation time as shown in Fig. 3d. The CNR drops to 50% of the maximum at a decorrelation time of ~200 μs, matching well with the period of the galvo mirror, 250 μs.

The ability of OCIS to form a focus pattern can be generalized to arbitrary intensity control through scattering media by modulating the light intensity through the optical channels. One notable scenario is the generation of a null energy spot at a target point. Unlike the display process to form an optical spot where we chose the bright speckles, here we inverted the measured logic signals to choose the dark speckles, which subsequently modulated the laser source as the synchronized galvo mirror scanned backward. As such, a null energy spot was observed on the time-integrated pattern with a PBR of $5.0 \times 10^{-2}$ and a CNR of −5.1 (Fig. 3e, h). To capture a control image, we randomly selected a subset of speckle patterns during display, and no null energy spot was observed (Fig. 3f, h). A quantitative derivation of the CNR and PBR of null energy spot patterns is included in Supplementary Note 1.

The process of speckle pattern selection and summation here shares similarity with the operational process of ghost imaging[16].

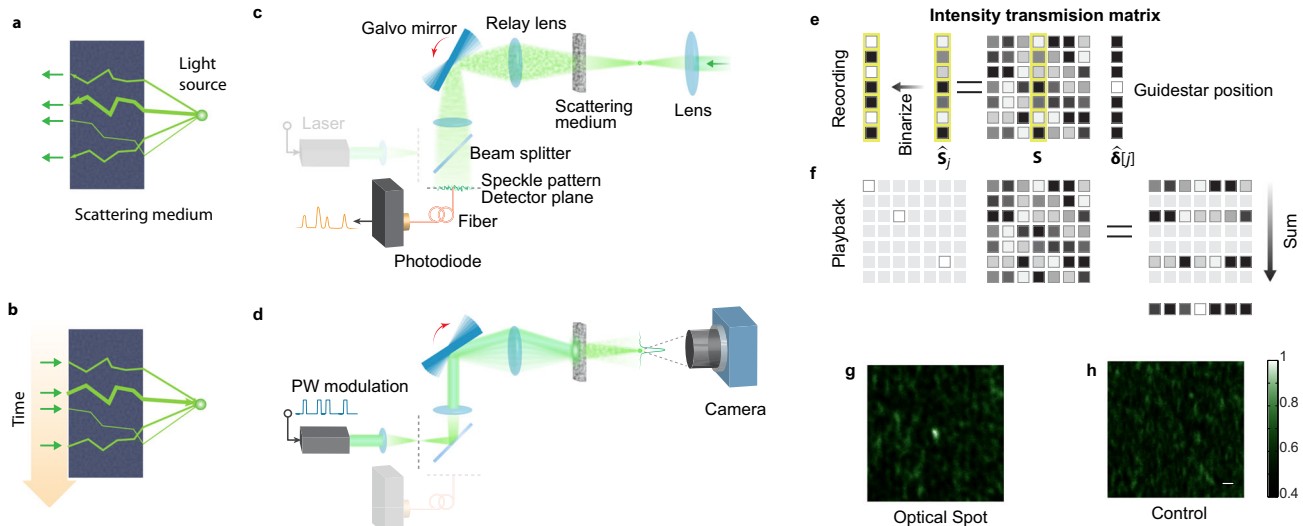

**Fig. 4 Optical intensity transposition. a, b** The principle of optical intensity transposition. **a** Light emitted from a coherent light source at the target plane traverses the optical channels to the input plane. By measuring the light intensity of the transmitted light as a function of space, we can obtain a spatial map of the optical channels. **b** By sending light back through the bright channels and linearly combining the transmitted light, we can obtain a focus pattern at the source location. **c, d** Experimental setup. Similar to optical phase conjugation, the retroreflecting process can be separated into two steps. **c** Recording. A point source transmitted light through the scattering medium and a photodiode measures a one-dimensional speckle pattern during one galvo mirror sweep. **d** Playback. The measured signal is time-reversed and then modulates the laser on the input side as the galvo mirror scans backward. An optical spot is created at the position of the initial point source. **e, f** Matrix representation of optical intensity transposition. During recording **e**, a guidestar selects a column of the intensity transmission matrix as a time-varying intensity output, which is then binarized. During playback (**f**), the binarized signal is used to select the corresponding rows of the intensity matrix, whose sum leads to a bright optical spot at the guidestar position. **g, h** Experimental results. An optical spot pattern was imaged on a camera with optical intensity transposition (**g**), while no bright spot was observed in the control experiment where we disabled the synchronization between the playback sequence and the galvo mirror (**h**). Scale bar: 20 μm.

However, there are fundamental distinctions between these two methods. First, ghost imaging measures speckles in free space and is not related to the optical channel theory, while OCIS is a method to overcome or utilize disordered scattering. Second, ghost imaging reconstructs images computationally, while OCIS is able to physically form images (see Supplementary Methods—Image transmission through scattering media with OCIS).

**Optical intensity transposition.** The ability to measure light transmitted from a point through a scattering medium and to find an optical solution that can send light back through the scattering medium to the original point (e.g., Fig. 4a, b) is highly sought for practical applications. In combination with Guidestar techniques[3], this ability can potentially allow controlled concentrating of light energy within and information transmission through a scattering medium. To date, optical phase conjugation is the dominant wavefront-shaping approach to perform such a function[17–20]. As such, the phase conjugation operation has long been assumed to be vital for retroreflecting light through or inside scattering media.

OCIS provides a simple alternative approach to accomplish the same objective—we name this approach optical intensity transposition. Moreover, OCIS accomplishes this objective without requiring phase measurements or the use of phase conjugation (Fig. 4c, d). To better understand the approach, we refer back to the intensity transmission matrix theory. From this theory, we can see that the index set $C_{FB}$ or the target column vector of the intensity transmission matrix records the throughput of the optical channels between the input plane and target spot. Interestingly, the feedback-based OCIS method is not the only way to obtain this information about the optical channels. One can also utilize a point source on the target plane, which can be formulated as a delta column vector $\hat{\delta}[j]$ (Fig. 4e, right column;

the symbol ^ denotes column vector), to probe the target column of the intensity transmission matrix, that is, $\hat{S}_j = S\hat{\delta}[j]$, in which $\hat{S}_j$ is the $j$th column of the intensity transmission matrix $S$. In this case, we can directly obtain the target column $\hat{S}_j$ on the input plane (Fig. 4e) as the response of the delta function on the target plane. By measuring and thresholding this column vector, we are able to obtain the index set $C_{OIT}$ with the optical intensity transpose method, which is the same as the $C_{FB}$. Once we have the information about the optical channels, we can follow the same procedure as feedback-based OCIS to control light intensity through scattering media, e.g., forming an optical spot on the target plane (Fig. 4f), which is mathematically described in Eq. (2). Interestingly, we can also playback all the modes simultaneously provided that they are spatially incoherent. Similar to Eq. (3), the summation of the selected intensity patterns on the input plane, $a_{OIT} = \sum_{i \in C_{OIT}} A_i$, is the transpose of the binarized column vector $\hat{S}_j$, justifying the name of optical intensity transposition. As a comparison, optical phase conjugation plays back the conjugate transpose of the measured column of the complex transmission matrix.

We demonstrate this concept by the following experiment. Similar to optical phase conjugation, optical intensity transposition also starts with a coherent light source or a Guidestar point[3] on the target plane behind the scattering medium (Fig. 4c). The resulting optical pattern on the detector plane after scatter by the scattering sample carries important information about the scattering characteristics of the medium. Instead of recording the complex field or its phase map, here we only record the intensity information. In this case, we scan a galvo mirror placed at the Fourier plane of the detector plane to convert a spatial intensity pattern into a temporal intensity signal and record the signal using a photodetector (Fig. 4c). See

Methods and Supplementary Fig. 2b for more details on the setup. In the playback step, instead of using a spatial light modulator to display a conjugated optical wavefront, here we simply turn on the light source when the backward-scanning galvo mirror rotates to the positions where bright speckles were measured on the detector plane during recording. This process can be easily achieved by modulating the light source with the time-reversed signal as the galvo mirror scans backward (Fig. 4d). By integrating the playback pattern over the backward scanning period on a camera, we observe an optical spot on the camera (Fig. 4g). In other words, we "refocused" light through the scattering medium to the origin by simply reflecting the light back without the need to consider and manipulate the phase information. As a control experiment, we mismatched the timing between the phase of the galvo mirror and the modulation signal and captured a control pattern as shown in Fig. 4h.

It is worth noting that wavefront shaping with amplitude-only modulation such as with digital micromirror devices (DMD) can also include similar operations—intensity measurement, thresholding, and modulation[19]. However, amplitude modulation-based wavefront shaping still exploits interference between the modulating pixels and therefore, is still a phase-based approach. As phase and interference are involved, a reference beam, parallel spatial modulation, and finely tuned alignment are all necessary with DMD-based phase conjugation. The underlying principle of OCIS is based on the linear operation on the intensity instead of the complex field in wavefront shaping, including DMD-based phase conjugation.

**Secure optical information transmission through scattering media.** Disordered optical scattering scrambles the propagation directions of photons. In optical imaging, this effect broadens the point spread function and prevents optical information from being localized or resolved precisely through scattering media. In free-space optical communication, the spreading of the photons

due to optical scattering prevents the information from being confined or delivered specifically and privately. In this case, the optical scatterers act as tiny "beam splitters" that duplicate and broadcast the optical information to the public.

Conventionally, a separate key is used to secure information transmission. In this case, a private channel is first established to allow the communication participants to share the key. The key is then used to encrypt the information to be transmitted in a public channel. Only the receivers with the key can decode the encrypted information. Of particular interest to mention here are the previous works on using optical approaches to generate random keys[21–24].

Recently, optical wavefront-shaping-based approaches have been demonstrated to address the nonspecific transmission of optical information due to optical scattering. Instead of using a separate digital key for encryption, this method takes advantage of the random scattering itself to "encrypt" to optical information[25]. In this case, the scrambled optical information due to optical scattering can only be recovered by the measured complex transmission matrix of the scattering.

Here, we demonstrate one potential use of OCIS in secure optical communication through scattering media. There are two main advantages of using OCIS in secure communication with the involvement of optical scattering. First, a separate random key generator and a private channel are not required although they are compatible with OCIS for an additional layer of security. Second, prior measurement of the complex transmission matrix is not required, which is important for remote communications where complex field measurement is challenging.

To understand the principle of OCIS-based information transmission through scattering media, we consider a scenario where person A (Alice) and person B (Bob) would like to communicate in a non-line-of-sight environment such as through a fog, turbid water, or around the corners. The scattering medium can also take the form of a multimode fiber. Similar to the optical intensity transposition described in the previous section, Alice will first illuminate the scattering medium with a point source to establish the channels (Fig. 5a). Bob will then measure the

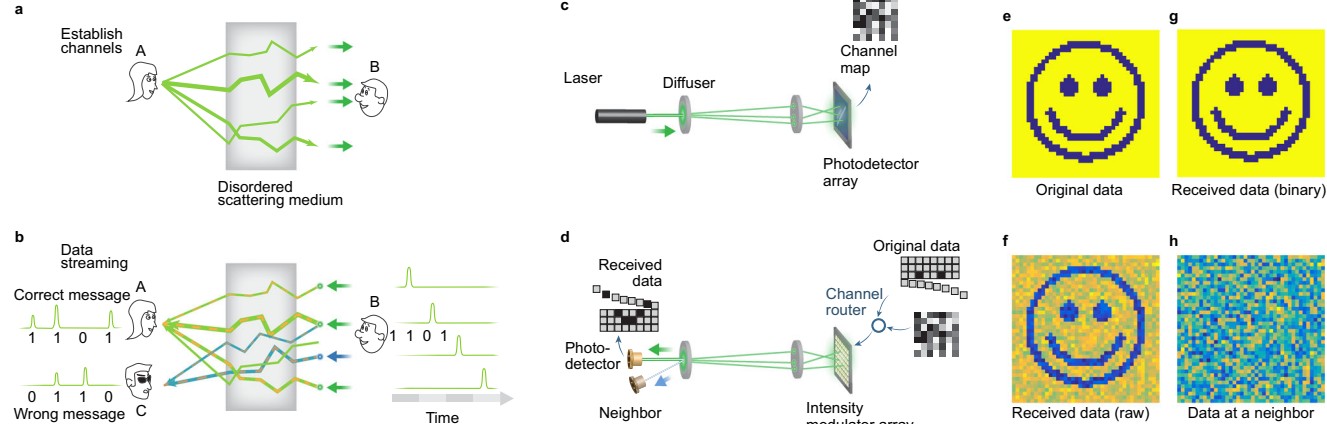

**Fig. 5 Secure communication with OCIS. a**, **b** Conceptual illustration. OCIS enables secure free-space optical communications between communication parties Alice and Bob. **a** Alice establishes optical channels between Alice and Bob by sending a laser pulse through the scattering medium. Bob measures the resultant speckle intensity pattern on the remote end to reveal the optical channels between Alice and Bob. **b** To send a binary message to Alice, Bob streams spatially incoherent optical pulses through different channels of the scattering medium, e.g., logical ones to randomly selected bright channels and logical zeros to randomly selected dark channels. As a result, Alice receives a matched message, while a third person Chuck receives a random message because the channels between Bob and Chuck are uncorrelated with those between Alice and Bob. **c**, **d** Experimental demonstration of the OCIS-based free-space secure communication. **c** A laser beam transmits through a local and a remote diffuser. A photodetector array at the remote end measures the resultant speckle pattern as the map of the optical channels. **d** An intensity modulator array is used to route the optical pulses to different channels based on the message and the measured channel map. Two photodetectors on the local side record the returned optical pulses. One of the photodetectors is conjugated to the laser, while the other one is placed elsewhere. **e**–**h** Experimental results. **e** Original binary data. **f** Raw data received by the conjugated photodetector. **g** Binarized data of (**f**). **h** Raw data received by the neighboring photodetector.

transmitted speckle pattern on a camera. Each bright speckle spot on the camera will represent a bright channel through the scattering medium back to Alice's initial point source. In other words, if Bob places a point source at that bright speckle location, Alice will receive a bright speckle. The opposite is true for the dark speckle spots on Bob's camera. Placing a point source at one of these dark points will cause Alice to receive a dark speckle. The relationship allows Bob to send a '1' bit (or '0' bit) by injecting photons to bright channels (or dark channels) as shown in Fig. 5b. As long as Bob only uses each channel once, the security of the communication channel would be preserved. An eavesdropper Chuck detects light elsewhere will not be able to glean useful information as he will receive a speckle pattern that is uncorrelated with Alice and Bob's (see Supplementary Note 3 for more details).

Figure 5c, d shows the schematic of OCIS-based free-space secure communications between communication parties Alice and Bob. Each of them used a ground glass diffuser as the scattering medium. During the channel establishment phase, a camera on Bob's side was used to record the speckle intensity pattern as a channel map (c). Then a DMD, which was pixel-to-pixel matched with the camera, was used to select bright or dark channels, depending on the logic values of the message to be transmitted (d). To enhance SNR, we combined multiple channels simultaneously to transmit one-bit data. At the same time, the photodetector on Alice's side will receive a binary intensity sequence that matches the original data. It should be noted that to avoid optical field interference between these channels, the light field on the DMD is spatially incoherent (see Methods and Supplementary Fig. 2c for more details). The experimental results are shown in Fig. 5e–h. The original data from Bob are a two-dimensional image (Fig. 5e), which was transmitted row by row to Alice. Each bit is either logic 1 or logic 0, which corresponds to a focus or a null pattern described in the section "Feedback-based OCIS". The CNR is ~3.7 for the focus pattern and $\sim -1.5$ for the null pattern. Upon reception by the photodetector on Alice's side, the data stream was reconstructed to an image (Fig. 5f), whose binarized version (Fig. 5g) matches the original data. In contrast, a photodetector that measured one speckle grain of the intercepted light, which mimics an eavesdropper Chuck, received a random sequence (Fig. 5h) that is uncorrelated with the original data.

To further enhance security, Alice can additionally shuffle the scattering medium (e.g., by rotating the diffuser) before her emitter and receiver to refresh the optical channel map intermittently. Effectively, secret and ever-changing channels are created between the sender and the targeted receiver, and information-only streams within the channels. In addition, OCIS provides a physical layer of encryption that is highly compatible and complementary to conventional digital key encryption. In the case where the digital key is hacked, OCIS serves as another line of defense, and vice versa. More details about the analysis on possible attacks and applicability of OCIS-based secure communications can be found in Supplementary Note 3.

## Discussion

OCIS opens up a door to control light through scattering media. While wavefront shaping proactively controls the interference of light between optical channels, OCIS takes advantage of the interference naturally formed by the scattering media. The concept of OCIS extends our understanding of controlling light through scattering media. In addition, it comes with a number of important features.

First, the OCIS implementation does not require direct or indirect wavefront measurement and modulation, instead OCIS

functions by using only linear intensity operations. At the system architecture level, OCIS's data flow structure is simple. It is this simplicity that allows our OCIS experiment to demonstrate focus (CNR > 12) generation in 250 µs experimentally. A theoretical speed analysis is provided in Supplementary Note 2. We anticipate that the system speed can be significantly boosted by using much faster intensity modulation schemes such as amplitude modulation of diode lasers and using faster scanners such as swept-source lasers. We further note that speed improvements are generally coupled with diminished photon budgets. An analysis of OCIS performance in the regime where shot noise becomes considerable is provided in Supplementary Note 1. One significant disadvantage of OCIS versus standard wavefront shaping is that the PBR and CNR enhancements are weaker functions of the number of controllable modes than those of wavefront shaping. OCIS's PBR is preset by choosing a threshold, and, as such, access to more optical modes ($N$) only allows us to pick a higher threshold. Optimizing PBR this way would yield ln($N$) peak improvements as derived in Supplementary Note 1. In comparison, PBR scales as $N$ for wavefront shaping. This indicates that wavefront shaping should generally outperform OCIS in light-starved scenarios (e.g., single-photon regime) or when the goal is to achieve intensity enhancement. When photon budget is not a limiting factor and the goal is to recover information from random scattering, CNR which measures the peak to the noise fluctuations (rather than the background DC value) is actually a better gauge of the nominal SNR performance. By this measure, the CNR of OCIS scales as $\sqrt{N}$ while wavefront shaping scales as $N$.

Second, OCIS is intrinsically capable of displaying negative patterns. The generation of negative patterns through scattering media is, in principle, possible with wavefront shaping but highly impractical as the negative pattern created would be difficult to be distinguished from naturally occurring null points in the speckled background. In contrast, because OCIS directly operates on an intensity basis instead of controlling interference, the background formed by OCIS exhibits a much less pronounced spatial variation. Similar to the formation of a bright spot, the controllable dark spot here also carries information through scattering media. In the secure communication application, the use of null spots to carry information is useful because it improves the overall single-to-noise characteristics of the method.

Third, since OCIS directly operates on intensity, it is useful in some applications where phase measurement is difficult to achieve. In many practical cases such as optical communications through fog, cloud, turbid water, walls, or biological tissue, it is difficult to implement a reference beam. The demonstrated secure information streaming through scattering media is an application enabled by the reference-beam-free feature of OCIS.

Last but not least, OCIS can be implemented with spectral channels. In this case, different optical wavelengths provide different optical channels because the scattering properties are wavelength-dependent. Instead of scanning through different spatial modes of the sample, one can also scan through the spectral modes to measure the throughput of the optical channels as a function of wavelength.

With its ability to confine optical information locally and specifically, OCIS can be used for imaging through scattering media as we have demonstrated in Supplementary Experiment and Supplementary Fig. 8. With the same principle, OCIS can also be used in secure information transmission where it helps prevent optical information from spreading globally in optical information transmission through scattering media. While we have demonstrated a free-space communication scenario, the OCIS can potentially be used with multimode fibers to secure information during transmission. OCIS provides physical

encryption that does not require the use of a digital key yet it is also compatible with conventional digital key encryption and thus provides an additional layer of security. This flexibility can potentially enable the use of OCIS in a wide range of optical information transmission applications.

In conclusion, we would like to close by noting that this is an initial report of OCIS for controlling light transmission through scattering media. While OCIS has its shortcomings in certain aspects as discussed above, it also provides advantages and insights from alternative perspectives. This concept can potentially inspire other systems and applications (e.g., working with a Guidestar mechanism for deep-tissue imaging[26]) to overcome or leverage disordered scattering in the future.

## Methods

**Experimental setups.** The optical setup of feedback-based OCIS is shown in Supplementary Fig. 2a. A collimated CW laser beam (532-nm wavelength, CrystaLaser Inc.) was intensity-modulated by an acousto-optic modulator (AOM, 100 MHz, IntraAction Corp.) by taking the first order of the diffracted beam. The modulated beam was then scanned by a galvo mirror (CRS 4 kHz, Cambridge Technology), which was imaged onto the surface of a ground glass diffuser (DG10-120, Thorlabs) through a 4-f relay system (L1 and L2). The light intensity on the surface of the diffuser was ~20 mW. Another 4-f system (L3 and L4) magnified the speckle to match the core diameter of the fiber. A photomultiplier tube (PMT, H7422, Hamamatsu) was used to measure the speckle intensity, and the output signal was sent to an analog comparator (LM361N, Texas Instruments). An FPGA board (Cyclone 2, Altera) that was synchronized with the galvo mirror received and processed the output signals from the comparator. The output signals from the FPGA controlled an electronic switch (ZASWA-2-50DR+, Mini-circuits) to modulate the amplitude of the carrier (100 MHz) to the AOM. A camera (GX1920, Allied Vision) was placed on the conjugate plane of the fiber to observe the optical patterns.

The optical setup of optical intensity transpose is shown in Supplementary Fig. 2b. During the recording process (Supplementary Fig. 2b1), lens L3 created an optical spot behind the ground glass diffuser and the optical spot was conjugated with the camera by a 4-f system (L3 and L4). The light from the optical spot was then scattered by the diffuser and the PMT recorded the intensity on the Fourier plane of the galvo mirror, which was conjugated to the surface of the diffuser. During the playback process (Supplementary Fig. 2b2), the collimated laser beam that was aligned to be conjugated to the fiber end was modulated by the AOM when the galvo mirror was scanning. In the same way as feedback-based OCIS, the FPGA received the binarized signals from the comparator and output the signals to control the AOM for OCIS.

The optical setup for realizing secure information transmission through scattering media is shown in Supplementary Fig. 2c. We used a ground glass diffuser, the same as the one used in feedback-based OCIS demonstration, as the scattering medium on each side. The camera measured a speckle pattern after light transmitted through the scattering medium. During data streaming, we randomly selected ~300 subchannels (corresponding to ~300 speckles) to form a channel. The light-intensity modulation was realized with a DMD system (Discovery 4100, Texas Instruments). To assure linear-intensity operation as described in the intensity transmission matrix theory, the DMD modulates spatially incoherent light, which was scattered by a rotating diffuser in front of the coherent laser source.

## Data availability

The data that support the plots within this paper and other findings of this study are available from this shared folder https://drive.google.com/drive/folders/1_2m6gNBWMu1LSqg1wB0JOHpUkIj88Y6c?usp=sharing or from the corresponding authors upon reasonable request.

## Code availability

The code that supports the experiments of this study is available from this shared folder https://drive.google.com/drive/folders/1_2m6gNBWMu1LSqg1wB0JOHpUkIj88Y6c?usp=sharing or from the corresponding authors upon reasonable request.

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

## Acknowledgements

We thank Dr. Yan Liu for the helpful discussions, Ruizhi Cao for the assistance in the experiments, and Dr. Joshua Brake for assistance in the experiments and improvements of the paper. This work is supported by the National Institutes of Health BRAIN Initiative Awards (U01NS090577).

## Author contributions

H.R. conceived the ideas. H.R., J.X., and C.Y. developed the ideas and designed the experiments. J.X. conducted the optical experiments with assistance from H.R. H.R., J.X., and C.Y. analyzed the data. All authors contributed to the preparation of the paper.

## Competing interests

The authors declare no competing interests.
