## [Peer Review File · Nature Communications]

Reviewer #1 (Remarks to the Author):

This is an intriguing paper where the authors rely on incoherent superpositions of speckles to transfer information, instead of the more conventional approach of coherent superpositions. In general, the performance in terms of CNR and PBR is not as good as with coherent superpositions, but there are a few windows of opportunity where incoherent superposition might win. For one, it is arguably easier to implement than coherent superposition, and faster. But the major drawback is that it is not light efficient, and only works well when a lot of power is available. It is not clear to me what impact this technique will have because of its limited range of applications. Perhaps the secure communication application is the most promising. But this leads me to another reservation, which is that this paper is quite long and sprawling. As it is, the messaging is diluted and unclear. There are many different experiments and setups. Moreover, some of the most interesting results are relegated to the Supplement. My feeling is that this work would be better served as two papers instead of one. The first paper could include the OCIS concept including intensity transposition and the imaging section in the Supplement. The second paper could then be devoted to secure communication only. This would help avoid the vacillations between pitching for biological and non-biological applications. It would also allow more room for experimental data, which, as it stands, is somewhat limited for a NC publication (for example, do the experiments actually match with predicted CNR and PBR in different intensity regimes? Can non-laser-based imaging be performed?).

Some minor comments:

- 1) In Fig 2, the "modes" are defined by their positions, but in Fig 3 they are defined by their angles. I understand they are roughly equivalent, but why the change?
- 2) Regarding the imaging experiment in Fig. S8, the two spots on the object are coherently illuminated during direct observation, not incoherently as in the principle of OCIS. As a result, I would expect the image at the observer plane to be speckly. This won't be apparent with an object as simple as two spots, but it will be apparent with a more extended object. In fact, a more interesting demonstration might be to use an incoherent object (an LED array? fluorescence?)
- 3) The text is well written for the most part, but there are some sections where it suddenly becomes uneven.
- 4) The null energy point doesn't seem that interesting/useful to me and takes up more space than it deserves, diluting the messaging further.

Reviewer #2 (Remarks to the Author):

Review of manuscript "Optical information transmission through complex scattering media with optical-channel-based intensity streaming" by Ruan, Xu, and Yang

In this manuscript, the authors experimentally demonstrate a focusing mechanism for light passing through scattering medium, and perform measurements on light passing through a glass diffuser and through a multimode fiber. In their method, Ruan et al achieve the focusing by adding the intensity of the bright speckles on a chosen spot for different directions of the impinging light beam using an open shutter on the detector (accumulating the electronic signal in time). The coherence of the light source is thus irrelevant for the enhancement they achieve, unlike the previously demonstrated wave-front shaping method for which constructive interference is a determining factor for achieving a high focus intensity relative to the background.

Technically, this paper achieves a relatively fast focusing speed by using programmable electronics connected to a fast scanning galvo mirror. To my knowledge, this combinations of instruments have not been reported before as a viable method for focusing through complex scattering media. The reported results are complete and I see no major barrier in reproducing these results in other

labs (except for the potential waste of other researchers' time because the code for analysis and the code for programming the FPGA are not made open source).

The authors also dedicate a major part of their manuscript to build support for the "fundamental" (authors choice of word on page 14) distinction of their method with previously reported focusing schemes that also use intensity-only modulation, for example Binary amplitude modulation. Such claims are repeated several times in various sections of the manuscript, from abstract to conclusions. Here I mention and challenge the fairness of some of these claims that are highlighted in the abstract, and I invite the authors to revisit their comparison with other methods based on a demonstrable performance instead of semantics.

Claims in the abstract:

The authors state there is a wide spread assumption that "knowledge of the optical phase is a necessity for optical control through scattering media". I do not recognize such an assumption in the field as a majority of articles on feedback-controlled focusing through scattering media, including the original 2007 letter by Vellekoop and Mosk actually used only the intensity, and sometimes even the incoherent fluorescence intensity, for optimization.

Many of the schemes that use digital micromirror devices (e.g. Akbulut et al, Optics Express 2011) also use (binary) intensity modulation (and do not manipulate phase of a channel) as a feedback signal. No interferometry or phase measurement is required for focusing with those methods. Yes, those methods benefit from the coherence of the laser source but they also demonstrate a very large enhancement as it scales with the N , number of control channels, instead of \sqrt{N} of the current report.

The author could explain to the readers why a sacrifice of slower scaling is necessary or how it is advantageous?

The authors state that their choice of temporally accumulating the intensities, instead of allowing constructive interference, "simplifies" the system, but I could not find an argument in the manuscript that explains how a FPGA- controlled fast scanning mirror is simpler to set up than an spatial light modulator, an adaptive mirror, or a DMD array.

The other claim that this algorithm "speeds up" the focusing process, without mentioning a figure of merit is also hard to verify. I am wondering for example, how a macroscopic scanning mirror can eventually perform faster than close to megahertz refresh rates of 1D spatial light modulators, as reported by Tzang, Piestun, et al in Nature Photonics 2019?

To conclude, while the technical side of this paper is an interesting contribution to the field of light control through scattering media, the argumentation for novelty and comparison with the literature is somewhat misleading for the readers who are not familiar with the literature of this field.

Sanli Faez, Utrecht University

Reviewer #3 (Remarks to the Author):

The manuscript by Ruan et al. present a framework to control the transmission of light through a scattering media relying only on intensity measurements and intensity modulation. In contrast to standard wavefront shaping strategies their technique is completely oblivious to the phase of the light field. They present experiments on feed-back based focusing and focusing by optical intensity transposition as well as a communication protocol that introduces a physical layer of security

The manuscript is clear and well written. The results are novel and interesting and the amount of different experiments performed (including the SM) is impressive. Also, the theoretical analysis of the achievable focus contrasts and intensities compared to the background is sound and well

explained.

The main concern I have is the question of actual usefulness in an imaging scenario. The authors show that the technique is fast and relatively easy to implement as only intensity modulations are necessary. However, the focus intensities realised are not impressive and the intrinsic background is large. I wonder, given an obtained focus can be scanned within the memory effect range, how big of a signal would one expect when imaging multiple fluorescent targets? Wouldn't the signal quickly vanish in the background if the number of targets is increased. For non-linear guidestars, would feedback based OCIS converge to a single focus among multiple targets?

In general, I'm open to recommending the work for publication in Nature Communications if all raised points are addressed (see also below). The work will probably spark interest in the field and might encourage thinking outside the wavefront shaping paradigm.

Several more specific comments, questions and critiques are detailed below:

(1) The whole concept of OCIS crucially relies on the use of incoherent light or the temporal separation of transmission channels in case of coherent illumination. This is neither mentioned in the abstract nor the introduction, only at end of page 5. Although for someone in the field it is quickly clear that the proposed technique could not work in the standard case of continuous coherent illumination, this restriction needs to be addressed more explicitly.

(2) The authors use the terms 'open' ('closed') channels to refer to collections of paths that interfere constructively (destructively) at a specific target position at other side of the scattering medium. This might lead to confusion as, in the context of scattering media, 'open' ('closed') channels generally refers to specific eigenmodes that are nearly perfectly transmitting (not transmitting at all). These are rooted in mesoscopic physics and are not connected to the channels employed in this work. Although the authors briefly mention this fact (top of page 5), different terms could be used to avoid confusion (e.g. 'bright' and 'dark' channels, as used later in the text). Also, could it be that reference 13 was meant to refer to Vellekoop and Mosk, Phys. Rev. Lett. 101, 120601 (2008) instead of the Optics Express?

(3) In support of their secure information transition protocol, the authors discuss a possible man-in-the-middle attack in the Supplementary Note 3. They show that a third party intercepting the light patterns in the middle of the scattering medium cannot easily retrieve the transmitted information from correlating the intensity patterns, given that the number of modes is large enough. This is quite an important point to prove for a secure communications protocol. Here, I was wondering if the third party could use a different strategy. What if they would select a very bright speckle grain in the initial pulse sent by Alice and then observe the intensity only in that grain when Bob sends his message (doing basically the same as Alice does to decode the message). Would this lead to a more favourable SNR/CNR for the interception, and how does it compare to Alice's CNR?

(4) Concerning the direct imaging scheme presented in Supplementary Figure S8 the authors state that they use a laser source modulated by an SLM to create the imaged object of two neighbouring spots. Why does the coherence of the laser not hinder the use of OCIS in that situation? The two spots are turned on and off such that an image forms from the averaged scanned speckles on the camera, but they are always on at the same time, aren't they? I would have expected this application to also necessitate the use of incoherent light. Further, I would have thought that due to the necessary incoherence the speckle on the camera loses its contrast and that imaging more complex structures other than two or three spots would not be possible?

(5) In the caption of Supplementary Figure S2 there is no reference to the individual subfigures c1, c2 and d.

(6) A minor remark on the the intensity transmission matrix: The same concept was recently used in Boniface et al., arXiv:2003.04255 (2020) where it seems to appear naturally in the transmission of incoherent fluoresces signals through a scattering medium.

Reviewer #1 (Remarks to the Author):

This is an intriguing paper where the authors rely on incoherent superpositions of speckles to transfer information, instead of the more conventional approach of coherent superpositions. In general, the performance in terms of CNR and PBR is not as good as with coherent superpositions, but there are a few windows of opportunity where incoherent superposition might win.

We thank the reviewer for the insightful summary.

For one, it is arguably easier to implement than coherent superposition, and faster. But the major drawback is that it is not light efficient, and only works well when a lot of power is available. It is not clear to me what impact this technique will have because of its limited range of applications. Perhaps the secure communication application is the most promising. But this leads me to another reservation, which is that this paper is quite long and sprawling. As it is, the messaging is diluted and unclear. There are many different experiments and setups. Moreover, some of the most interesting results are relegated to the Supplement. My feeling is that this work would be better served as two papers instead of one. The first paper could include the OCIS concept including intensity transposition and the imaging section in the Supplement. The second paper could then be devoted to secure communication only. This would help avoid the vacillations between pitching for biological and non-biological applications. It would also allow more room for experimental data, which, as it stands, is somewhat limited for a NC publication (for example, do the experiments actually match with predicted CNR and PBR in different intensity regimes? Can non-laser-based imaging be performed?).

We thank the reviewer for the suggestion of splitting the manuscript into two. We have considered this suggestion extensively. There are several reasons for which we present our work in the current form.

First, OCIS is a new type of strategy for controlling light through scattering media, that is quite unlike conventional wavefront shaping. We feel it is important to demonstrate how OCIS as an alternative method maps onto the key capabilities of wavefront shaping, i.e. iterative wavefront optimization, digital optical phase conjugation, image transmission through transmission matrix, and direct imaging through scattering media. By drawing these connections, we hope to provide a clear and complete picture of OCIS to the readers from multiple perspectives.

Second, while we could demonstrate a fast imaging application after explaining the concept, this feature is marginal and less effective to distinguish OCIS from wavefront shaping. The secure information transmission, on the other hand, is a relatively new and conceptual demonstration where OCIS have a clearer advantage and can thus deliver a larger impact. As the first paper of OCIS, we feel that it's more impactful to deliver the concept and the enabled application together.

While we ultimately decided to keep the work in one paper, we also understand the reviewer's concern on the length and structure of the presentation. To improve on this aspect, in the revised manuscript, we have removed a large portion of the reporting on the null energy point demonstration among others to make the manuscript concise. Please find the second and third last paragraphs of the Feedback based OCIS section for the deletions. In addition, we have also highlighted the structure of the work in the revised abstract.

Some minor comments:

1) In Fig 2, the “modes” are defined by their positions, but in Fig 3 they are defined by their angles. I understand they are roughly equivalent, but why the change?

We thank the reviewer for pointing out this discrepancy. This confusion is likely because we choose to use ‘position scanning’ to more clearly describe the scanning task in a pictorial form as shown in Fig. 2, while we employ angular scanning in the specific experiment for practical reasons. To avoid misunderstanding, we have added more details to clarify. Please see the last sentence in the first paragraph of the Feedback based OCIS section - “It should be noted that in addition to position scanning, other approaches such as angular scanning and wavelength swept can also be used to excite different channels of a scattering medium.”

2) Regarding the imaging experiment in Fig. S8, the two spots on the object are coherently illuminated during direct observation, not incoherently as in the principle of OCIS. As a result, I would expect the image at the observer plane to be speckly. This won’t be apparent with an object as simple as two spots, but it will be apparent with a more extended object. In fact, a more interesting demonstration might be to use an incoherent object (an LED array? fluorescence?)

It’s true that the two spots are coherently illuminated and interfere on the imaging plane at each optical pulse. Between optical pulses, however, the relative phase between the two light fields from the two spots is random. Therefore, statistically the interference has minimal impact on the mean intensity of the spots on the imaging plane over the integration of many pulses. On the other hand, the interference between the two spots within an optical pulse period will lead to a speckle background of both doubled mean intensity and standard deviation. Therefore, the background fluctuation increases by N when the number of spots increases by N . The number of spots that can be imaged also depends on the number of controllable modes or operation time as shown in Supplementary Fig. S6, where CNR scales up with square root of number of controllable modes.

The reviewer raised a good point regarding the use of incoherent objects to demonstrate the direct imaging method of OCIS. Since OCIS requires speckle patterns at the imaging plane, an ideal source should be temporally coherent but spatially incoherent. This type of light source can be achieved by using a rotating diffuser to scramble coherent light. Alternatively, a narrow band LED or fluorescence may also provide decent CNR. Different from using spatially and temporally coherent illumination, in this case, the light from N objects add up incoherently and the fluctuation increases by \sqrt{N} .

Since this is the first demonstration of OCIS’s direct imaging feature, we tried to keep our setup simple. Nevertheless, the reviewer raised a good point and we should clarify the light source requirements and characteristics. In the revised manuscript, we have added a discussion of light source in the last paragraph of the Supplementary Methods.

3) The text is well written for the most part, but there are some sections where it suddenly becomes uneven.

We have gone through our manuscript and made substantial revision. We hope our manuscript reads better this time.

4) The null energy point doesn’t seem that interesting/useful to me and takes up more space than it deserves, diluting the messaging further.

We thank the reviewer for the comment. The null energy point is arguably a unique aspect of OCIS. We feel that reporting it will help readers better understand OCIS and perhaps inspire novel applications. On the other hand, we also understand the concern from the reviewer since the manuscript is already long and covers many experiments. In the revised manuscript, we have removed a significant portion of the reporting on the null energy point from the Feedback based OCIS section. Please find the second and the third last paragraphs of this section for the deletion. In addition, we have also removed the mentioning of this feature in the abstract.

Reviewer #2 (Remarks to the Author):

Review of manuscript "Optical information transmission through complex scattering media with optical-channel-based intensity streaming" by Ruan, Xu, and Yang

In this manuscript, the authors experimentally demonstrate a focusing mechanism for light passing through scattering medium, and perform measurements on light passing through a glass diffuser and through a multimode fiber. In their method, Ruan et al achieve the focusing by adding the intensity of the bright speckles on a chosen spot for different directions of the impinging light beam using an open shutter on the detector (accumulating the electronic signal in time). The coherence of the light source is thus irrelevant for the enhancement they achieve, unlike the previously demonstrated wave-front shaping method for which constructive interference is a determining factor for achieving a high focus intensity relative to the background.

Technically, this paper achieves a relatively fast focusing speed by using programmable electronics connected to a fast scanning galvo mirror. To my knowledge, this combinations of instruments have not been reported before as a viable method for focusing through complex scattering media. The reported results are complete and I see no major barrier in reproducing these results in other labs (except for the potential waste of other researchers' time because the code for analysis and the code for programming the FPGA are not made open source).

We thank the reviewer for the insightful summary and comments on the novelty and soundness of the work. We have made our codes open source. Please find the Data and Code Availability section in the revised manuscript for the link to the code.

The authors also dedicate a major part of their manuscript to build support for the "fundamental" (authors choice of word on page 14) distinction of their method with previously reported focusing schemes that also use intensity-only modulation, for example Binary amplitude modulation. Such claims are repeated several times in various sections of the manuscript, from abstract to conclusions. Here I mention and challenge the fairness of some of these claims that are highlighted in the abstract, and I invite the authors to revisit their comparison with other methods based on a demonstrable performance instead of semantics.

Claims in the abstract:

The authors state there is a wide spread assumption that "knowledge of the optical phase is a necessity for optical control through scattering media". I do not recognize such an assumption in the field as a majority of articles on feedback-controlled focusing through scattering media, including the original 2007 letter by Vellekoop and Mosk actually used only the intensity, and sometimes even the incoherent fluorescence intensity, for optimization. Many of the schemes that

use digital micromirror devices (e.g. Akbulut et al, Optics Express 2011) also use (binary) intensity modulation (and do not manipulate phase of a channel) as a feedback signal. No interferometry or phase measurement is required for focusing with those methods. Yes, those methods benefit from the coherence of the laser source but they also demonstrate a very large enhancement as it scales with the N, number of control channels, instead of \sqrt{N} of the current report. The author could explain to the readers why a sacrifice of slower scaling is necessary or how it is advantageous?

We agree with the reviewer that some wavefront shaping techniques measure the intensity information and modulate intensity. However, the key enabling factor in these techniques is still the manipulation of interference between different optical modes. In the original 2007 letter by Vellekoop and Mosk, for example, the intensity is used as a feedback for the spatial light modulator to tune the phase of each pixel and to achieve constructive interference between pixels. In essence, they are quantifying phase relationships through intensity measurement since conventional detectors can only measure intensity directly. Similarly, for wavefront shaping schemes that involve digital micromirror devices (DMDs), the DMDs are used to control the interference of light through different paths by turning on only pixels for which light fields are in phase. Therefore, we would consider these techniques as still requiring knowledge of optical phase, albeit indirectly in some of these cases. In contrast, the operation of OCIS does not require cross-interference between light associated with each pixel (or channel).

We appreciate the points raised by the reviewer on the potential confusion in articulating the difference between OCIS and wavefront shaping in terms of practical measurement and modulation. We carefully went through the manuscript and edited the wording on the description on phase and intensity measurement and modulation. In brief, instead of attributing the difference on intensity vs phase measurement and modulation, we use the term active interference to distinguish these two classes of technique across the manuscript.

The main focus of this manuscript is the new concept of OCIS. Since both wavefront shaping and OCIS have various specific implementations, e.g. feedback based focusing and optical phase conjugation, the comparison between them is nuanced. For example, in terms of the secure information transmission application, OCIS is likely advantageous. As this is the first report on OCIS, we focused on reporting on the principle and various properties of OCIS. The experiments reported here are designed to best illustrate the operating principle of OCIS, rather than optimized for speed or other performance benchmarks.

The reviewer's suggestion for more comparative analysis is well received. We have revised the Discuss section to better describe the pros and cons of OCIS.

The authors state that their choice of temporally accumulating the intensities, instead of allowing constructive interference, "simplifies" the system, but I could not find an argument in the manuscript that explains how a FPGA- controlled fast scanning mirror is simpler to set up than an spatial light modulator, an adaptive mirror, or a DMD array.

The other claim that this algorithm "speeds up" the focusing process, without mentioning a figure of merit is also hard to verify. I am wondering for example, how a macroscopic scanning mirror can eventually perform faster than close to megahertz refresh rates of 1D spatial light modulators, as reported by Tzang, Plestun, et al in Nature Photonics 2019?

The simplicity of the system is reflected at a system architecture level. In principle, the electronic part of OCIS only requires a comparator, a data buffer, and an intensity modulator – electronically simple. We anticipate this simplicity can lead to a higher response speed and other tangible benefits. However, to do a fair comparison with wavefront shaping methods, we would need to implement an optimized OCIS system and do a direct comparison, which is beyond the scope of this proof-of-concept paper. In view of this, while we have added more details to describe the system architecture, we have removed the statements about simplifying the system across the manuscript. Please find the second paragraph of the discussion section for the added description on the system architecture.

Comparison of performance speed in the context of wavefront shaping is nuanced and complex. The reference mentioned by the reviewer is a good illustration of that complexity. Specifically, the megahertz refresh rates claimed by Tzang et al. can be easily misinterpreted. In wavefront shaping, we desire the ability to respond quickly to changes in the scattering medium. Therefore, the response speed is a more practical metric than the refresh rate of the modulation device. As reported in their paper, it takes 2.4 ms to form a focus of CNR of ~44. The process of forming a wavefront-shaped focus involves making interference measurements, processing the data and feeding the correct input to the modulation device. Simply having a fast modulation device does not imply a fast response speed if other upstream processes are the bottleneck. In contrast, OCIS only takes 0.25 ms to form a focus of CNR of ~13. This improvement in response speed can be directly attributed to the OCIS's system architecture simplicity (e.g. no need for each mode to interfere with three phase references as required in the referred work), which improves the overall data flow. Speed comparison is complex because, for practical applications, we generally want to consider the achievable CNR in the calculus (in the above comparison, OCIS clearly underperformed on CNR). We believe such a comparison between OCIS and wavefront shaping is best done in a subsequent paper with an optimized OCIS system built for speed. To address the reviewer's concern at this point, we have removed mentions of 'speed up' and have made no claims about speed advantages in the revised manuscript.

To conclude, while the technical side of this paper is an interesting contribution to the field of light control through scattering media, the argumentation for novelty and comparison with the literature is somewhat misleading for the readers who are not familiar with the literature of this field.

We thank the reviewer again for helping to improve the quality of the manuscript. We have revised the manuscript based on the reviewers' comments.

Reviewer #3 (Remarks to the Author):

The manuscript by Ruan et al. present a framework to control the transmission of light through a scattering media relying only on intensity measurements and intensity modulation. In contrast to standard wavefront shaping strategies their technique is completely oblivious to the phase of the light field. They present experiments on feed-back based focusing and focusing by optical intensity transposition as well as a communication protocol that introduces a physical layer of security

The manuscript is clear and well written. The results are novel and interesting and the amount of different experiments performed (including the SM) is impressive. Also, the theoretical analysis of the achievable focus contrasts and intensities compared to the background is sound and well explained.

We thank the reviewer for the insightful summary and the compliment on the novelty and quality of the work.

The main concern I have is the question of actual usefulness in an imaging scenario. The authors show that the technique is fast and relatively easy to implement as only intensity modulations are necessary. However, the focus intensities realised are not impressive and the intrinsic background is large. I wonder, given an obtained focus can be scanned within the memory effect range, how big of a signal would one expect when imaging multiple fluorescent targets? Wouldn't the signal quickly vanish in the background if the number of targets is increased. For non-linear guidestars, would feedback based OCIS converge to a single focus among multiple targets?

The reviewer raised good questions about the potential performance of OCIS in fluorescence imaging. The fluorescence imaging signal depends on the number of optical modes that OCIS scans and displays. Supplementary Fig. S7a shows that the contrast to noise ratio (CNR) of an OCIS focal pattern increases by the square root of operation time. If the sample decorrelates at 1 ms and the operation time is also set to ~1 ms, the theoretical CNR can reach ~28 as shown in Supplementary Fig. S7b. As the number of the fluorescent targets increases to N , the image CNR scales down by a factor of $1/\sqrt{N}$. Therefore, given a sample of 1 ms decorrelation time, the image CNR drops to 1 when OCIS images ~784 (28^2) fluorescent targets in principle. The image CNR or the number of fluorescent targets can further increase by repeating and accumulating the measurements.

Using non-linear guidestars with OCIS is an interesting idea. Since OCIS relies on the superposition of the speckle intensity in space over time, it would be difficult for OCIS, in its current form, to take advantage of nonlinear effects. However, we do note that OCIS is new and there may be other OCIS variants that can better leverage nonlinear effects.

In general, I'm open to recommending the work for publication in Nature Communications if all raised points are addressed (see also below). The work will probably spark interest in the field and might encourage thinking outside the wavefront shaping paradigm.

We thank the reviewer for acknowledging that OCIS will likely spark interest and encourage new way of thinking about optical control through scattering medium.

Several more specific comments, questions and critiques are detailed below:

(1) The whole concept of OCIS crucially relies on the use of incoherent light or the temporal separation of transmission channels in case of coherent illumination. This is neither mentioned in the abstract nor the introduction, only at end of page 5. Although for someone in the field it is quickly clear that the proposed technique could not work in the standard case of continuous coherent illumination, this restriction needs to be addressed more explicitly.

We thank the reviewer for the suggestion. We have added a description about the property of the light source in the fourth paragraph of the introduction - "Since the phase relationship between these channels is not measured, a spatially incoherent light source or temporal separation of the coherent light transmissions is used to achieve linear superposition of the photons from different channels."

(2) The authors use the terms 'open' ('closed') channels to refer to collections of paths that interfere constructively (destructively) at a specific target position at other side of the scattering medium. This might lead to confusion as, in the context of scattering media, 'open' ('closed')

channels generally refers to specific eigenmodes that are nearly perfectly transmitting (not transmitting at all). These are rooted in mesoscopic physics and are not connected to the channels employed in this work. Although the authors briefly mention this fact (top of page 5), different terms could be used to avoid confusion (e.g. 'bright' and 'dark' channels, as used later in the text). Also, could it be that reference 13 was meant to refer to Vellekoop and Mosk, Phys. Rev. Lett. 101, 120601 (2008) instead of the Optics Express?

We thank the reviewer for the comments and suggestions regarding channel terminology. We agree that 'bright' and 'dark' channels are better choices in this context and have made this change in the manuscript.

For reference 13, yes, Vellekoop and Mosk, Phys. Rev. Lett. 101, 120601 (2008) is actually what we meant. We appreciate the careful review by the reviewer.

(3) In support of their secure information transition protocol, the authors discuss a possible man-in-the-middle attack in the Supplementary Note 3. They show that a third party intercepting the light patterns in the middle of the scattering medium cannot easily retrieve the transmitted information from correlating the intensity patterns, given that the number of modes is large enough. This is quite an important point to prove for a secure communications protocol. Here, I was wondering if the third party could use a different strategy. What if they would select a very bright speckle grain in the initial pulse sent by Alice and then observe the intensity only in that grain when Bob sends his message (doing basically the same as Alice does to decode the message). Would this lead to a more favourable SNR/CNR for the interception, and how does it compare to Alice's CNR?

The 'brightest speckle' approach proposed by the reviewer for eavesdropping is an interesting strategy. To estimate the feasibility of this approach, we need to calculate the power contribution of the brightest speckle to a speckle that Bob measures. Mathematically, the expected power ratio between the brightest speckle and the overall power is approximately $(1+\ln(M))/M$, where M is the total number of modes. If $M=1e6$ for example, the power ratio is $\sim 1.5e-5$, which is also the expected power contribution from the brightest speckle to a speckle measured by Bob. Therefore, the connection between a bright speckle that Bob measures and the brightest speckle at the intercepting plane is extremely weak. As a result, observing the brightest speckle should not provide sufficient information to decode the message that Bob sends. In contrast, Alice should always be able to receive the information from Bob through the established channel.

(4) Concerning the direct imaging scheme presented in Supplementary Figure S8 the authors state that they use a laser source modulated by an SLM to create the imaged object of two neighbouring spots. Why does the coherence of the laser not hinder the use of OCIS in that situation? The two spots are turned on and off such that an image forms from the averaged scanned speckles on the camera, but they are always on at the same time, aren't they? I would have expected this application to also necessitate the use of incoherent light. Further, I would have thought that due to the necessary incoherence the speckle on the camera loses its contrast and that imaging more complex structures other than two or three spots would not be possible?

In the direct imaging scheme, it is true that the two spots are on at the same time and that they interfere with each other within each optical pulse period. Between optical pulses, however, the relative phase between the optical fields from these spots changes. Therefore, over a large number of pulse integration, the mean intensity of optical spots on the imaging plane is

independent of the number of spots. On the other hand, both the mean intensity and standard deviation of the speckle background increase by a factor of two due to the use of coherent illumination.

The reviewer raised a good point on using an incoherent source for a complex object. Since a speckle pattern from each spot is required, an ideal light source would be temporally coherent and spatially incoherent, e.g. a coherent source with its phase scrambled by a rotating diffuser. In this case, the standard deviation of the background is proportional to the square root of the number of spots. Therefore, if the number of controllable modes is reasonably high (Supplementary Fig. S7 and Note 2), OCIS should be able to image more than a few points with this type of light source.

Since this is the first demonstration of OCIS and its application in direct image transmission, we kept our experiments simple. Nevertheless, the reviewer raised a good point on the impact of coherence state. In the revised manuscript, we have added a paragraph in the Supplementary Methods to discuss this point.

(5) In the caption of Supplementary Figure S2 there is no reference to the individual subfigures c1, c2 and d.

We thank the reviewer for the careful review. We have added the captions for the subfigures.

(6) A minor remark on the the intensity transmission matrix: The same concept was recently used in Boniface et al., arXiv:2003.04255 (2020) where it seems to appear naturally in the transmission of incoherent fluoresces signals through a scattering medium.

We thank the reviewer for referring to this paper. We have cited this paper in the third paragraph of the Principles section in the revised manuscript.

REVIEWER COMMENTS

Reviewer #1 (Remarks to the Author):

The authors have addressed most of my comments satisfactorily (as well as the other reviewer comments). I recommend publication. I have only one parting comment related to Fig S8, and to the new last paragraph in the supplement. I think the requirement of temporal coherence only applies to large spots. If the spots are small, they need not be temporally coherent.

Reviewer #2 (Remarks to the Author):

The authors have responded thoroughly and diligently to my comments. I am convinced that this revised presentation of the context and results of this new experimental scheme can be of great interest for the community. I therefore can recommend the publication of the manuscript in its revised form.

I also would like express my gratitude to the authors for making the operation and analysis codes available.

Dr. Sanli Faez

Utrecht University

Reviewer #3 (Remarks to the Author):

The authors sufficiently addressed most of my concerns. Some minor points are followed up on below. I am still not fully sure if this technique will find many applications but it is surely an interesting approach to the problem light scattering.

.) Although the authors acknowledge that reference 13 is supposed to refer to Vellekoop et al. PRL 101, 120601 (2008) it still refers to the Optics Express of the same year!?

[SEP.] Where I am not sure I fully understood the authors response is in the case of the role of coherence in the direct imaging scenario discussed in the Supplementary. The authors write that “between optical pulses, however, the relative phase between the optical fields from these spots changes”. With this, do they mean that the speckles originating from the two sources do interfere at each pulse in the image plane, however, the background field from upper spot that interferes with the light from the bright channel of the lower spot has a random phase and therefore on average the stronger bright spot survives. This would make sense to me. Is the (perceived) comparatively enhanced speckled-ness of the final image maybe a result of this?

Reviewer #1 (Remarks to the Author):

The authors have addressed most of my comments satisfactorily (as well as the other reviewer comments). I recommend publication. I have only one parting comment related to Fig S8, and to the new last paragraph in the supplement. I think the requirement of temporal coherence only applies to large spots. If the spots are small, they need not be temporally coherent.

We thank the reviewer for recommending publication and for the comment. We agree that temporal coherence is not required if the spots are small. In this case, a narrow band source should be sufficient. We have modified the statement in the last paragraph of the supplementary information – “To obtain a higher CNR for a complex object, we can consider using a temporally coherent (or narrow band) and spatially incoherent light source, e.g. using a rotating diffuser to scramble the phase of a coherent source.”

Reviewer #2 (Remarks to the Author):

The authors have responded thoroughly and diligently to my comments. I am convinced that this revised presentation of the context and results of this new experimental scheme can be of great interest for the community. I therefore can recommend the publication of the manuscript in its revised form.

I also would like express my gratitude to the authors for making the operation and analysis codes available.

We thank the reviewer for recommending publication of the work.

Reviewer #3 (Remarks to the Author):

The authors sufficiently addressed most of my concerns. Some minor points are followed up on below. I am still not fully sure if this technique will find many applications but it is surely an interesting approach to the problem light scattering.

.) Although the authors acknowledge that reference 13 is supposed to refer to Vellekoop et al. PRL 101, 120601 (2008) it still refers to the Optics Express of the same year!?

We thank the reviewer again for the careful review. We have updated the references and corrected the mistake in the revised manuscript.

.) Where I am not sure I fully understood the authors response is in the case of the role of coherence in the direct imaging scenario discussed in the Supplementary. The authors write that “between optical pulses, however, the relative phase between the optical fields from these spots changes”. With this, do they mean that the speckles originating from the two sources do interfere at each pulse in the image plane, however, the background field from upper spot that interferes with the light from the bright channel of the lower spot has a random phase and therefore on average the stronger bright spot survives. This would make sense to me. Is the (perceived) comparatively enhanced speckled-ness of the final image maybe a result of this?

We agree with the reviewer's interpretation. We had elaborated on the role of coherence in the last paragraph based on the reviewer's explanation – "Although the light field of a focus on the image plane interferes with the background light field from the other spot, their phase relationship is random between pulses. Therefore, the expected intensity of the focus after integrating many pulses remains the same as the case of one spot. However, both the mean intensity and standard deviation of the background are doubled due to the interference between the two spots." The use of the coherent source indeed enhanced the speckled-ness of the final image.